

# Assessing agriculture's vulnerability to drought in European pre-Alpine regions

Ruth Stephan[1], Stefano Terzi[2,3,4], Mathilde Erfurt[1], Silvia Cocuccioni[2], Kerstin Stahl[1], and Marc Zebisch[2]

[1]Environmental Hydrological Systems, University of Freiburg, Freiburg i. Br., 79085, Germany
[2]Institute for Earth Observation, Eurac Research, Viale Druso 1, 39100, Bolzano, Italy
[3]Center for Global Mountain Safeguard Research, Eurac Research, Viale Druso 1, 39100, Bolzano, Italy
[4]United Nations University Institute for Environment and Human Security (UNU-EHS), Platz der Vereinten Nationen 1, 53113 Bonn, Germany

**Correspondence:** Ruth Stephan (ruth.stephan@hydrology.uni-freiburg.de)

**Abstract.** Droughts are natural hazards that lead to severe impacts in the agricultural sector. Mountain regions are thought to have abundant water, but have experienced unprecedented drought conditions as climate change is affecting their environments more rapidly than other places. The effect radiates by reducing water availability well beyond the mountains' geographical locations. This study aims to improve the understanding of agriculture's vulnerability to drought in Europe's pre-Alpine region,

considering two case studies that have been severely impacted in the past. We applied a mixed-method approach combining the knowledge of regional experts with quantitative data analyses in order to define region-specific vulnerability based on experts' identified factors. We implemented two aggregation methods by combining the vulnerability factors that could be supported with subregional data. Whereas the equal weighting method combines all factors with the same weight, the expert weighting method combines the factors with varying weight based on expert opinion. These two methods resulted in vulnerability maps

with the expert weighting showing in general higher vulnerability, and partly relocating the medium and lower vulnerabilities to other subregions within the case study regions. In general, the experts confirmed the resulting subregions with higher vulnerability. They also acknowledged the value of mapping vulnerability by adopting different aggregation methods confirming that this can serve as a sensitivity analysis. The identified factors contributing most to the regions' vulnerability point to the potential of adaptation strategies decreasing the agriculture's vulnerability to drought that could enable better preparedness.

Apart from region-specific differences, in both study regions the presence of irrigation infrastructure and soil texture are among the most important conditions that could be managed to some extent in order to decrease the regions' vulnerability. Throughout the analyses, the study benefited from the exchange with the experts by getting an in-depth understanding of the regional context with feedback-relations between the factors contributing to vulnerability. Qualitative narratives provided during the semi-structured interviews supported a better characterization of local vulnerability conditions and helped to better identify

quantitative indicators as proxies to describe the factors. Thus, we recommend to apply this mixed-method approach to close the gap between science and practitioners.





# 1 Introduction

Past and recent droughts have led to severe environmental, social and economic impacts in many regions of the world. The combination of climate change exacerbation and the arising pressures on water demand from socio-economic activities affect

the intensity and frequency of drought conditions (van Loon et al., 2016). This is particularly relevant in mountain regions where climate change effects are occurring more rapidly than in other places with consequences on their water tower role and water provision to downstream areas (Beniston and Stoffel, 2014; Immerzeel et al., 2020; Terzi et al., 2021). Recent drought events highlighted the vulnerability of the European Alps and areas dependent on water from the Alps such as pre-alpine regions to unexpected conditions of reduced water availability (Hanel et al., 2018; Laaha et al., 2017). Stephan et al. (2021)

showed that within the European Alps a wide range of drought impacts occurred in different socio-economic sectors, with agriculture and the public water supply most impacted.

Drought impacts are triggered by the natural hazard itself, such as the intensity, duration, frequency and extent of water deficits, but local exposure and vulnerability characteristics shape them (Hagenlocher et al., 2019). While the drought hazard components have been investigated and a set of indices are already established and available to describe hydroclimatic vari-

ations in terms of precipitation (e.g. SPI) and evapotranspiration (e.g. SPEI) in the Alpine region (e.g. Haslinger and Blöschl (2017)), the characterization of drought vulnerability and exposure still remains a challenge. Multiple conceptual frameworks of vulnerability in the context of natural hazards assessments have been developed (Birkmann et al., 2013; González Tánago et al., 2016). The Intergovernmental Panel on Climate Change (IPCC) provides the risk concept definition, where exposure refers to 'the presence of people, livelihoods, species or ecosystems, environmental functions, services, and resources, infras-

tructure, or economic, social, or cultural assets in places and settings that could be adversely affected' by a drought hazard' and vulnerability refers to the 'the propensity or predisposition to be adversely affected' (IPCC, p. 1048) due to the system's sensitivity or susceptibility combined with a lack of short-term coping capacity and long-term adaptive capacity. In case of drought risk assessments, an example of a short-term coping capacity is the existence of an irrigation system to reduce agricultural impacts, while the development of an agricultural system increasing the water use efficiency refers to a long-term adaptive

capacity. Although exposure and vulnerability are internationally recognised as important drivers of drought risk processes and final impacts, their operational assessment is still discussed and in development. In particular, drought vulnerability studies for areas such as the heterogeneous European Alpine and pre-alpine region are rare. So far studies exist on the aspects of vulnerability in forest growth and for impacts on pasture (e.g. Hartl-Meier et al. (2014)). Melkonyan (2014) Melkonyan (2014) carried out a study to assess socio-economic vulnerability of the agricultural sector in the mountainous region of Armenia.

Moreover, various approaches have been applied often relying on either quantitative or qualitative data and hence covering only specific aspects of drought risk processes. The selection of the underlying approach often depends on the study's spatial scale, as well as on the data availability. Most studies that adopt a quantitative approach focus on large scale (e.g. national or global level, Meza et al. (2020); Carrão et al. (2016)), where datasets on socio-economic conditions are freely available. Additionally, data on past drought impacts are also often only available on a large scale, but necessary for an effective validation

of the assessment. Regional datasets on socio-economic conditions are indeed often lacking, leading to an underrepresentation





or omission of regional risk conditions. Therefore, regional or local studies often follow a qualitative approach to assess vulnerability, e.g., by involving the local communities in the process of understanding the vulnerability dimensions (Ayantunde et al., 2015; Birhanu et al., 2017; Martin et al., 2016). The results of these two approaches have rarely been combined or compared.

For these reasons, this study considers the Impact Chains (IC), a mixed-method approach recognized as a valuable methodology to integrate both quantitative and qualitative information into the description of the hazard, exposure and vulnerability components for advancing the assessments of drought risk conditions (Schneiderbauer et al., 2020; Zebisch et al., 2021). The IC provides a guideline to systematically identify, select and assess relevant factors involved in risk processes through the combination of quantitative data with information coming from local experts and stakeholders (GIZ, 2017). This process is

particularly important for evaluations of the vulnerability and exposure components since no standard set of factors exists to identify and characterise vulnerability to drought in agriculture.

Besides the identification of vulnerability factors, expert knowledge is often used to weigh different factors in the mapping of an overall vulnerability index (Zebisch et al., 2021). This is often adopted when certain factors are perceived to be more important than others and thus have a greater (or lesser) influence on the overall vulnerability. In the context of impact chains,

equal weighting is usually applied more often than other weighting methods. Moreover, a comparison of different weighting methods in order to analyse the effect of such choice on the final vulnerability assessment, has rarely been done.

This study applies a vulnerability assessment approach in two case studies located in the European pre-Alpine region. The case study regions 'Thurgau' in Switzerland and 'Podravska' in Slovenia have experienced severe drought impacts in recent years and need to improve their resilience (Zorn and Hrvatin (2015); Komac et al. (2019), DROUGHT-CH). The overarching

objective is to systematically identify vulnerability factors linked to agricultural production that potentially lead to agricultural impacts during a drought to assess how they vary spatially.

Subordinate methodological research questions are:

– To what degree can we characterise vulnerability to drought in the two case studies by combining experts' opinions with the openly accessible data?

– How sensitive is the final vulnerability assessment to different weighting methods for the vulnerability factors?

## 2 Case study areas

We selected the two study regions 'Thurgau' in Switzerland and 'Podravska' in Slovenia due to the increasing number of reported drought impacts in agriculture in both regions, despite their proximity to the water-rich conditions in the European mountainous regions (Stephan et al., 2021). Most impacts archived in the Alpine Drought Impact report Inventory (EDII$_{ALPS}$)

report consequences on agriculture and livestock farming highlighting the agricultural sector of both regions vulnerable to drought in the past. According to EDII$_{ALPS}$, in Thurgau the most reported impacts occurred in 2015 and 2018, whereas in Podravska most impacts occurred in 2003 and 2017. In Thurgau, reports claimed a reduced productivity of annual and permanent





crop cultivation and shortages of feed and water for livestock as the majority of the regions' drought impacts. For instance
EDII$_{ALPS}$ reports a comment of a local farmer from Stettfurt about the considerable harvest loss: 'Production was down by

30 to 40 %. Especially iceberg lettuce had to be irrigated, because they were sensitive to the drought due to their small root

system. Irrigation had to be done even at night, so water from the Stettfurt public utility company had to be used in part, which

was much more expensive than the water from the Lauche River that he usually used'. In Podravska the reports are clearly

dominated by reduced productivity of annual crop cultivation, often with yield losses ≥ 30%. For example, the Drought Man-

agement Centre for Southeastern Europe stated in 2017, 'Agricultural drought [...] worsened over June and July. Most affected

were regions of northeastern and southern half of Slovenia where maize completely stopped. [...] yield was reduced by 30-50

%. Hay production was seriously affected as well.' (DMCSEE, 2017).

Both the Thurgau and Podravska regions are characterised by extensive areas of agricultural land (Fig. 1) that make them par-

ticularly exposed to drought conditions, while showing different societal and economic conditions influencing the vulnerability

of the agricultural sector. Moreover, they are also part of the Alpine Drought Observatory (ADO, https://www.alpine-space.

org/projects/ado), an Interreg project aiming to improve the understanding of drought processes and impacts towards increasing

levels of drought preparedness through monitoring. Thurgau is known for its agriculturally shaped landscape reaching from

specialty crops on the coast of Lake Constance to high elevation pastures (https://www.landschaftsqualitaet-tg.ch/). This study

region covers 991.77 km$^2$, 54 % of which are covered by agricultural land (Fig. 1, CLC, 2018), with a total of 2531 farms in

2019. The main agricultural type (51 %) consists of natural meadows and pastures, followed by cropland (33 %), managed

meadows (9 %), and vegetables, fruit, vines, berries (6.5 %) (LID, 2007). Thurgau had 279,547 inhabitants in 2019 (SFSO,

2022) resulting in a population density of 324 inhabitants per km$^2$. Podravska's landscape varies from hilly lowlands mainly

along the river Drava up to 1517 masl in the western mountainous part. The region covers an area of 2170 km$^2$, more than

double the size of Thurgau. In 2019, the region had 324,875 inhabitants and thus a lower population density (149.7 inhabitants

per km$^2$) than Thurgau ('population' from SURS, 2022). In Podravska more than 30 % of the total area is utilised as agricul-

tural land with a total of 10,990 agricultural holdings in 2016, and almost one third of these are located in mountainuous areas

('agriculture, forestry and fishery' from SURS, 2022).

## 3    Data and methods

This study applies a mixed-method approach based on qualitative and quantitative information for the assessment of the agri-

culture's vulnerability to drought conditions in the two case studies regions Thurgau and Podravska. Following the guidelines

of the vulnerability sourcebook (Fritzsche et al., 2014) our study approach was based on an initial exploratory phase during

the ADO project meeting held online on the 24th September 2020. During the group discussion, experts from the two case

study areas provided a first regional overview of the drought issues through the creation of impact chains following the model

by Zebisch et al. (2021). For this study we then considered these impact chains as context for further discussions and refine-

ments focusing on the vulnerability factors and their characteristics. Starting from this context, the methodological approach

presented here is composed of five consequential phases for the assessment of vulnerability considering its spatial variabil-





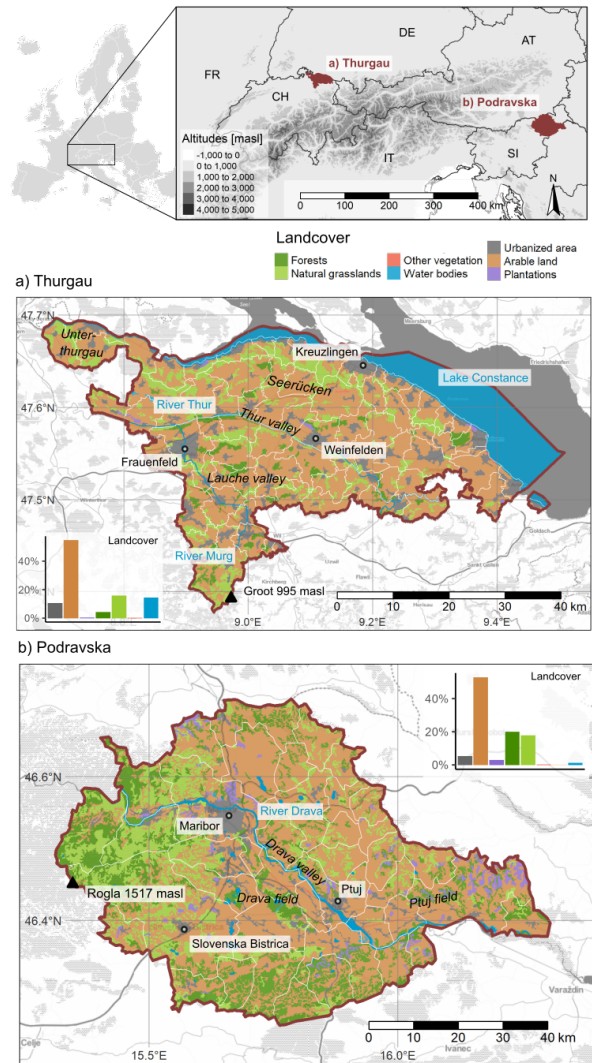

**Figure 1.** The two study regions a) Thurgau and b) Podravska within the European mountain region. The political border of the study regions is marked in red and the LAU2 regions are marked with white. The black labels present the most important cities and common known subareas (italic). The blue labels present the largest important rivers and lakes. The colours present the regional land cover adapted and modified by the Corine Land Cover (CLC, 2018). The land cover's share is shown in the histograms.

ity within both case study regions: (1) identification of vulnerability factors, (2) data acquisition and indicators selection, (3) pre-processing, normalisation and direction, (4) weighting and aggregating, and (5) participatory validation (Fig. 2).



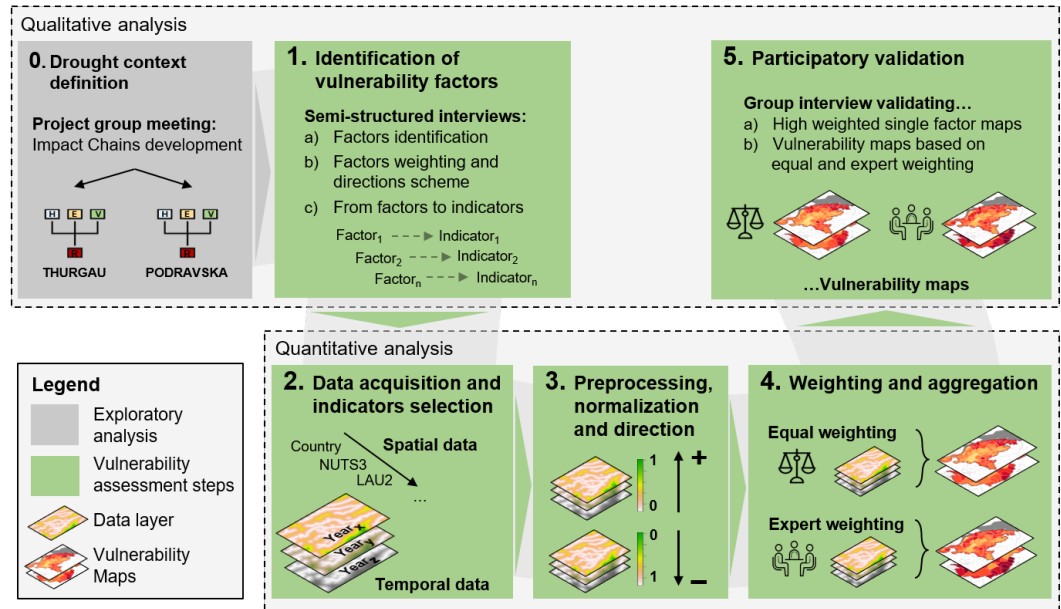

**Figure 2.** Conceptual overview of the methodological steps for the drought vulnerability assessment. The grey box represents the preliminary group discussion on agricultural drought while green boxes refer to the five consequential phases for a spatial assessment of drought vulnerability.

## 3.1 Identification of vulnerability factors

The set of factors used to describe vulnerability to drought in agriculture was generated through semi-structured interviews
with nine high-level experts identified as key people with extensive expertise and knowledge on the two case studies and they were held from the 24th August to 9th September 2021. Each discussion followed a flexible interactive structure (reported in S1) allowing to integrate the established questions with further information on the context and expertise from each participant. The interviews aimed to identify the perceived factors making the regions' agriculture vulnerable to drought and whether these factors increase or decrease the vulnerability. Further, the participants assigned an importance to each of the identified
factors from a low-medium-high scale of relevance. Finally, at the end of each interview we gathered information from the participants on which quantitative indicators from local biophysical and socio-economic datasets they deemed suitable to describe the spatial characteristics of the identified factors.

## 3.2 Data acquisition and indicator selection

In order to quantitatively describe the identified vulnerability factors, we scanned the datasets proposed by the interviewed
participants to select suitable indicators matching the meaning of each factor. In addition, we searched databases from authorities at regional (e.g Amt für Geoinformation Thurgau), national (e.g. Swiss Federal Statistical Office (SFSO), Statistical Office of the Republic of Slovenia (SURS)) to the European scale (e.g. European Environment Agency (EEA), European Soil Data





Centre (ESDAC)). Data availability and accessibility proved highly variable. If two different sources offered the same dataset to describe suitable indicators we prioritised the most recent and local dataset, as we aimed for current data with a spatial resolution as high as possible and at least with a subregional resolution of LAU1 regions.

### 3.3 Pre-processing, normalisation and direction

Depending on the type of data and data availability we gathered continuous and categorical data for both study regions. We transformed the categorical data into (a) continuous data by ordering the categories, or into (b) presence-absence data with the presence of the most important category according to the responses of the interview participants. In order to facilitate the analyses all available datasets were rasterized to grid data with the boundaries of Thurgau and Podravska and their region-specific projection. Thus, we gathered one raster layer for each indicator ($r_i$) representing the factor quantitatively, if subregional data is available.

In the next step, we directed $r_i$ according to the increasing and decreasing effect on the regions' vulnerability resulting from the interviews with the experts, similar to Meza et al. (2020). We transformed indicator layers with presence-absence data to values of '1', if presence has an increasing effect and to values of '0', if absence has an decreasing effect. Regarding the other indicator layers, we multiplied $r_i$ with '+1', if the indicator increases the vulnerability or we multiplied the values of $r_i$ with '-1', if the indicator decreases the vulnerability. Then, we normalised the directed indicator layers to account for different value ranges and units. Both steps are applied as follows

$$r_{i_{d,norm}} \in [0,1] = r_i * d \in \{-1, 1\} * \frac{r_i - min_{r_i}}{max_{r_i} - min_{r_i}} \tag{1}$$

where $r_{i_{d,norm}}$ represents the directed and normalised indicator layer, $r_i$ represents the indicator layer with the original values from its datasource(s), $d$ represents the direction factor in order to assign an increasing '+1' or decreasing '-1' effect, $min_{r_i}$ represents the minimum value across $r_i$, $max_{r_i}$ represents the maximum value across $r_i$. That means $r_{i_{d,norm}}$ presents with '0' the lowest possible vulnerability according to the indicator layer $r_i$, and with '1' the highest possible vulnerability according to the indicator layer $r_i$. All other values are transformed in-between with increasing values representing increasing vulnerability.

### 3.4 Weighting and aggregation

We applied two different weighting methods to calculate the vulnerability index $V$: the equal-based weighting method ($V_{eq}$), and the expert-based weighting method ($V_{ex}$). For each method the assigned weights to the indicators sum up to 1, and thus can be seen as proportional weights dependent on the method. We aggregated the weighted indicators by summing them up. Practically, we added the weighted raster layers corresponding to the indicators for each method. According to Fig. 1 the agricultural land is very heterogeneously distributed, wherefore we masked the summed indicators with the region-specific agricultural land by setting to zero the indicators' values in non-agricultural areas. By doing so, we accounted for the density of agricultural areas over the whole area within each LAU2 region in the next step. We aggregated the summed indicators at





each LAU2 region within Thurgau and Podravska calculating mean values of vulnerability for each LAU2 region. The final
maps enable a comparison of the two weighting methods for both study regions. The above described weighting methods are
specified in the following.

For $V_{eq}$ we assigned to each indicator $i$ the same weight $w_{i_{eq}}$ as follows

$$V_{eq} = \sum i \in \{1, 2, ..., n\} * r_{i_{d,norm}} * w_{i_{eq}}, \text{ with } \sum i \in \{1, 2, ..., n\} * w_{i_{eq}} = 1 \tag{2}$$

$$V_{eq} = \sum i \in \{1, 2, ..., n\} * r_{i_{d,norm}} * \frac{1}{n} \tag{3}$$

where $n$ represents the total number of all factors, $r_{i_{d,norm}}$ represents the directed and normalised indicator layer (see. eq.
1), and $w_{i_{eq}}$ represents the weight of indicator $i$ that equals in this method to $\frac{1}{n}$. In eq. 3 we aggregated the weighted rasters by
summing them up to the equal-based vulnerability method $V_{eq}$.

To gather $V_{ex}$, we used the importance rating Impr by the interview participants from the low-medium-high scale and the
frequency, how often a factor was named. We weighted each corresponding indicator $i$ with the expert-weight $w_{i_{ex}}$ as follows

$$w_{i_{ex}} = \sum i \in \{1, 2, 3\} * Imp_r * \frac{p_{i,r}}{W_{tot}} \tag{4}$$

$$W_{tot} = \sum i \in \{1, 2, ..., n\} * \sum i \in \{1, 2, 3\} * Imp_r * p_{i,r} \tag{5}$$

where $w_{i_{ex}}$ represents the experts' weight of indicator $i$, Impr represents the importance rate on the low-medium-high scale
translated into the integers 1-2-3, $p_i$, $r$ represents the number of the experts who weighted indicator $i$ with the weight $r$.
$W_{tot}$ represents the total weight assigned to all indicators, and $n$ represents the total number of all indicators. Analoge to the
equal weighting method, we weighted each indicator layer $r_{i_{d,norm}}$ with the corresponding weight $w_{i_{ex}}$ and summed up these
weighted raster layers to calculate $V_{ex}$ as follows

$$V_{ex} = \sum i \in \{1, 2, ..., n\} * r_{i_{d,norm}} * w_{i_{ex}}, \text{ with } \sum i \in \{1, 2, ..., n\} * w_{i_{ex}} = 1 \tag{6}$$

### 3.5 Participatory validation

In order to validate the results, we conducted one group interview involving the previously consulted experts for each case
study. We held the group interviews between the 10[th] and 21[st] June 2022 following the same structure (reported in S2). Firstly,
we asked the experts to identify the most and least vulnerable subregions across Thurgau and Podravska based on their past
experiences in dealing with drought events. Secondly, we presented the maps displaying the most important vulnerability fac-
tors and asked their opinions on the differences displayed across the study regions. Thirdly, we presented the final (aggregated)
vulnerability maps according to the two weighting methods and we asked for feedback on both the most and least vulnerable
regions. Finally, we asked them to provide their opinion on the differences between the maps based on the two weighting
methods and which one provides a better description of real drought vulnerability conditions in agriculture.



# 4 Results

## 4.1 Factors and indicators describing the region specific vulnerability

During the semi-structured interviews the experts named in total 31 unique factors describing vulnerability to agricultural
drought impacts (see Table 1). The experts identified 10 common factors for both study regions, whereas they identified 6
factors solely for Thurgau and 13 factors solely for Podravska (see Table 1). The common factors were altitudes, distance
to large water bodies, share of drought resistant crop types, farm size, humus content, presence of irrigation infrastructure,
slope, soil texture, topsoil depth and water holding capacity. The unique factors for Thurgau were south facing area, share of
intensive livestock, share of pastures, political conservative vote, share of specialty crops and type of irrigation infrastructure,
whereas the unique factors for Podravska were distance to mountains, landscape diversity, water permits, accessibility to local
food market, farm diversification, intensity of farming, absence of drought policies, food price, agro-technical measures, clear
landownership, compensations, farmers' age and farmers' education. Thus, the identified factors describe various aspects of
vulnerability, such as geographic conditions (e.g. elevation, south facing), hydrological characteristics (e.g. distance to large
water bodies), soil characteristics (e.g. water holding capacity), agricultural practices (e.g. intensive farming), agricultural
infrastructure (e.g. irrigation infrastructure), farmers' education, and policies (e.g. compensations). In comparison, the experts
from Podravska claimed more and a wider range of factors describing the regions' vulnerability than the experts from Thurgau.

For both regions we found suitable indicators with subregional data from various sources to cover a majority of the identified
vulnerability factors (see Table 1, Fig. S1). However, we were not able to support with subregional data 6 out of 18 factors in
Thurgau and 11 out of 25 factors in Podravska. For both regions we used data from the European Environment Agency (EEA,
2022) and from the European Soil Data Centre (ESDAC, 2022). Regarding Thurgau, we used data from national sources as
the Swiss Federal Statistical Office (SFSO, 2022) and the Swiss Federal Office of Environment (FOEN, 2022), and data from
regional sources the Office for Geoinformation by Thurgau (Amt für Geoinformation Thurgau, 2022). Regarding Podravka, we
used data from the Statistical Office of the Republic of Slovenia (SURS, 2022) and from the INSPIRE Slovenian Data Portal
(INSPIRE, 2022).
Some factors could be easily described through indicators supported by data available at the subregional scale. In these cases
the factors, originally described by the experts, were fully represented by the selected indicators. For example, with the digital
elevation model by the EEA we could represent the factor slope with the indicator slope measured in radians. In addition, with
a combination of shapefiles by the EU-Hydro - River Network Database respectively (EU-Hydro) for large rivers, reservoirs
and lakes we could represent the factor distance to large water bodies by calculating this distance for each location in both case
studies. For other factors we were forced to select proxy indicators which were either more specific or only covering a partial
aspect of the overall concept provided by the experts during the interviews. For example, for the factor presence of irrigation
infrastructure we found information about the presence or absence of permanently irrigated agricultural land across both study
regions and used this as a representative indicator. For example, in Thurgau we used the indicator dominant soil texture with
classes from clay to sand to describe the factor soil texture as a whole, and we used the indicator livestock units for the factor





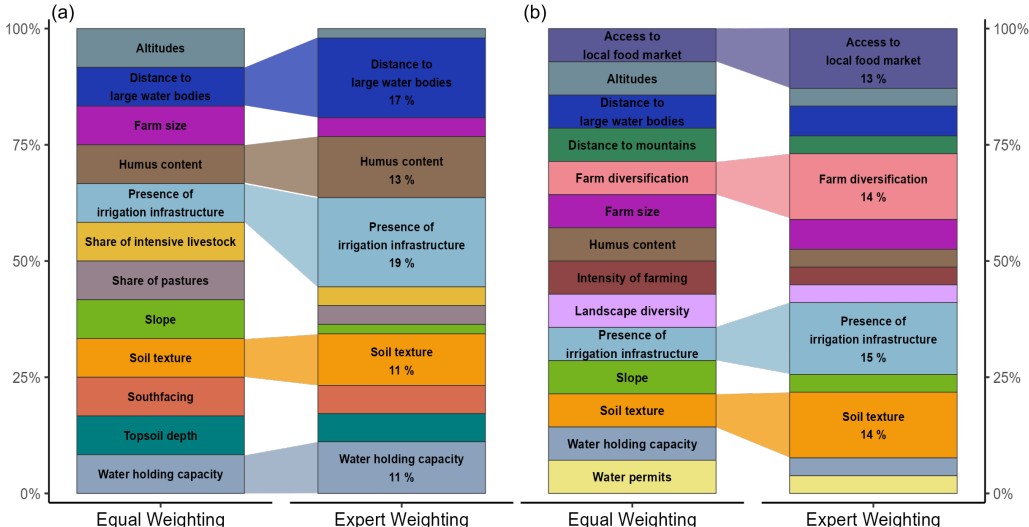

**Figure 3.** The identified vulnerability factors for (a) Thurgau and (b) Podravska that could be supported with subregional data. The size of each factor within the bars corresponds to the weight comparing the equal weighting method with the expert weighting method. For an interpretation of the references to colour in this figure legend, the reader is referred to the web version of this article.

share of intensive livestock. This way, we could support a majority of 12 factors in Thurgau and a majority of 14 factors in Podravska with data from indicators (see Table 1).

For some indicators the data is static over time, such as elevation. For indicators that show a temporal development we were able to find recent data, such as number of farms > 30 ha (from 2019 in Thurgau) and average utilised agricultural area per agricultural holding (from 2010 in Podravska). Regarding the available spatial resolution in Thurgau, we found shapefiles with different spatial resolution for five indicators, raster files with a resolution at least of 25 hectares for four indicators, and data for LAU1 regions for three indicators. In Podravska, we found raster files with a resolution of at least 25 hectares for six indicators, data for LAU2 regions for four indicators, with different spatial resolution for three indicators, and point data for one indicator. Most of the factors are represented by indicators with continuous data (respectively n = 6 and n = 10), followed by indicators with categorical data (n = 4 and n = 3) and by indicators with binary data (n = 2 and n = 1).

## 4.2 Vulnerability Thurgau

Thurgau's vulnerability is calculated with 12 factors for both the equal weighting method and the expert weighting method (see Fig. 3). The following five factors were attributed a greater importance by the experts and were therefore assigned with higher weights and accounting for a total of 71 % of the total vulnerability: distance to large water bodies, humus content, presence of irrigation infrastructure, soil texture and water holding capacity, whereas the other factors lost weight in the expert weighting method, specifically the factor slope.





**Table 1.** Set of factors the interview participants used to describe vulnerability to agricultural drought impacts and that could be supported with data in Thurgau and Podravska. The data source, data type, data range, the direction, the latest update and the spatial resolution of the data is shown.

**Thurgau**

| Factor[1] | Indicator | Direction | Latest update | Data type | Data source |
|---|---|---|---|---|---|
| Altitudes | Altitudes [masl] | ↘ | 2016 | Continuous, Raster | EU-DEM |
| Distance to large water bodies | Distance to large water bodies [km] | ↗ | 2020 | Continuous, Raster | EU-Hydro and FOEN |
| Farm size | No. of frams ≥ 30 ha [no.] | ↘ | 2019 | Continuous, Vector | SFSO |
| Humus content | Topsoil organic carbon content [%] | ↘ | 2006 | Classes, Vector | ESDAC |
| Presence of irrigation infrastructure | Permanently irrigated agricultural land | ↘ | 2018 | Binary, Raster | CLC |
| Share of intensive livestock* | Livestock units [LU] | ↗ | 2020 | Continuous, Vector | SFSO |
| Share of pastures* | No. of farms specialized for pasture farming [no.] | ↘ | 2020 | Continuous, Vector | SFSO |
| Slope | Slope [rad] | ↗ | 2014 | Continuous, Raster | CLC |
| Soil texture | Dominant soil texture | ↗ | 2005 | Binary, Vector | Amt für Geoinformation Thurgau |
| Southfacing area* | Southfacing | ↗ | 2005 | Binary, Vector | Amt für Geoinformation Thurgau |
| Topsoil depth | Dominant topsoil depth | ↗ | 2005 | Classes, Vector | Amt für Geoinformation Thurgau |
| Water holding capacity | Topsoil available water capacity [mm] | ↘ | 2006 | Classes, Vector | ESDAC |

**Podravska**

| Factor[1] | Indicator | Direction | Latest update | Data type | Data source |
|---|---|---|---|---|---|
| Access to local food market* | Agricultural holdings with main destination for sale [%] | ↘ | 2010 | Continuous, Vector | SURS |
| Altitudes | Altitudes [masl] | ↗ | 2017 | Continuous, Raster | INSPIRE |
| Distance to large water bodies | Distance to large water bodies [km] | ↗ | 2020 | Continuous, Raster | EU-Hydro and INSPIRE |
| Distance to mountains* | Distance to mountains [m] | ↗ | 2008 | Continuous, Raster | European Mountain Areas |
| Farm diversification* | Average no. of permanent beds per agricultural holding [no.] | ↘ | 2009/2010 | Continuous, Vector | SURS |
| Farm size | Average utilised agricultural area per agricultural holding [ha] | ↘ | 2010 | Continuous, Vector | SURS |
| Humus content | Topsoil organic carbon content [%] | ↘ | 2006 | Classes, Vector | ESDAC |
| Intensity of farming* | Average total yield per municipality [t] | ↗ | 2010/2020 | Continuous, Vector | SURS and INSPIRE |
| Landscape diversity* | Landscape diversity (Shannon's eveness index) | ↘ | 2018 | Continuous, Raster | CLC |
| Presence of irrigation infrastructure | Permanently irrigated agricultural land | ↘ | 2018 | Binary, Raster | CLC |
| Slope | Slope [rad] | ↗ | 2017 | Continuous, Raster | INSPIRE |
| Soil texture | Subsoil textural class | ↘ | 2006 | Classes, Vector | ESDAC |
| Water holding capacity | Topsoil available water capacity [mm] | ↘ | 2006 | Classes, Vector | ESDAC |
| Water permits* | Discharge from permits [l/y] | ↘ | 2012 | Continuous, Point data | INSPIRE |

[1] marked with '*' for factors only mentioned for either Thurgau or Podravska, the remaining were mentioned for both study regions.

Each mapped factor shows higher and lower vulnerabilities across the region according to the indicator values (see Fig. S2). The increasing and decreasing effect of the factor depends on the direction defined by the experts (see Table 1). The factor with the highest weight according to the experts is the presence of irrigation infrastructure which accounts for 19 % of the total vulnerability. Permanently irrigated land is relatively equally distributed across Thurgau. The LAU2 regions covered most



with irrigation infrastructure are in the central North of Thurgau (Homburg, Kemmental). Regions least covered with irrigation
      infrastructure are in the higher elevation-regions of the South (Fischingen) and east of the city of Constance and along the coast
      (Berlingen, Gottlieben). The second most important factor is distance to large water bodies accounting for 17 % of the total
      vulnerability. This mapped factor presents heterogeneous distances across the region, with lower vulnerability along the coast,
      the rivers Rhine, Thur and Murg, and around larger water reservoirs and lakes mostly located in the Northeast. Subsequently, the

highest vulnerabilities related to this factor occur in the LAU2 regions in-between the rivers and lakes (Homburg, Raperswilen,
      Wäldi) and in the mountainous South (Fischingen). The third most important factor in Thurgau is humus content accounting
      for 13 % of the total vulnerability. The best fitting indicator supporting this factor is called Topsoil organic carbon content [%]
      and offered by the European Soil Database (ESDAC, 2022). This categorical indicator presents higher carbon content and thus
      a lower vulnerability in the LAU2 regions located along the river Thur (Hüttlingen, Fellben-Wellhausen, Hohentannen). On

the other hand, most areas across Thurgau present very low carbon content leading to 21 LAU2 regions with carbon content <
      1 % spread across the study region. The fourth and fifth most important factors are soil texture and water holding capacity. For
      soil texture we used data from the Amt für Geoinformationen, which presents the dominant topsoil texture with five classes
      from clay to clay rich sand that we transformed in three classes with sand as the most vulnerable texture and clay as the
      least vulnerable texture, as described by the experts (see Table 1). Only a few areas spread across Thurgau are covered with

sandy soils and are thus most vulnerable (Basadingen-Schlattingen, Felben-Wellhausen, Lommis), whereas Thurgau is mostly
      covered by clay and silt decreasing the vulnerability. For the factor Water holding capacity we used the indicator Topsoil
      available water capacity from the European Soil Database (ESDAC, 2022). This indicator presents a very high capacity (>190
      mm) along the river Thur (Märstetten, Felben-Wellhausen, Wigoltingen). In contrast, lowest capacities (<99 mm) are presented
      for several LAU2 regions along the coastline and close to the river Rhine. Accordingly, the LAU2 regions in the South are less

vulnerable and the LAU2 regions in the North are more vulnerable with respect to this factor.

      In order to represent the overall vulnerability, we summed all factor raster layers and set to zero all non-agricultural areas in
      order to compute a mean value of vulnerability for each LAU2 region according to the equal ($V_{eq}$) and expert ($V_{ex}$) weighting
      methods (Fig. 4). To identify the agricultural areas we used regional data from the Office for Geoinformation by Thurgau (Amt
      für Geoinformation Thurgau, 2022).

The five highest Veq towards agricultural drought impacts occur in the central north and northeast part of Thurgau with $V_{eq}$
      > 0.25. The highest $V_{eq}$ is indicated for Raperswilen ($V_{eq}$ = 0.29), Berg ($V_{eq}$ = 0.27), Dozwil ($V_{eq}$ = 0.26), Wigoltingen ($V_{eq}$ =
      0.26), and Lommis ($V_{eq}$ = 0.25). All these LAU2 regions show a high share of agricultural land over the total region land, hence
      making them more exposed to possible drought impacts. In particular, although Raperswilen Wigoltingen and Lommis show
      a larger presence of irrigation infrastructure, many other factors, such as distance to large water bodies, humus content and

water holding capacity, contribute to an increased vulnerability. Differently from Raperswilen Wigoltingen and Lommis, the
      LAU2 region of Dozwil and Berg show lower presence of irrigation infrastructure, but low humus content and water holding
      capacity. The five lowest $V_{eq}$ are located in the central and southern part of Thurgau with $V_{eq}$ < 0.07. The lowest $V_{eq}$ values
      are reported for Fischingen ($V_{eq}$ = 0.05), Bichelsee-Balterswil ($V_{eq}$ = 0.05), Bettwiesen ($V_{eq}$ = 0.06), Wilen ($V_{eq}$ = 0.06),
      Wuppenau ($V_{eq}$ = 0.07). For all these regions, almost all underlying factors show moderate to low values besides presence of



irrigation infrastructure and share of intensive livestock having higher values. Additionally, a low share of agricultural land drives the low vulnerability. The highest vulnerability calculated with the expert-based method shows some differences in the spatial distribution of LAU2 regions with the northwest increasing in vulnerability. In particular, the region of Raperswilen is again ranked as first ($V_{ex}$ = 0.35), followed by Herdern ($V_{ex}$ = 0.31), Homburg ($V_{ex}$ = 0.30), Wäldi ($V_{ex}$ = 0.30), and Uesslingen-Buch ($V_{ex}$ = 0.28). For all these regions, the factors water holding capacity, humus content and distance to large

water bodies present high vulnerability, apart from Uesslingen-Buch with relatively short distance to the river Thur. In contrast, the factor soil texture presents these regions covered mostly by clay, and the factor presence of irrigation infrastructure displays almost all regions covered with irrigation systems, both decreasing vulnerability. However, the combined effect of a large share of agricultural land and visible patches of high vulnerability values across the factors distance to large water bodies, humus content and water holding capacity contribute to the final high vulnerability values. The LAU2 regions with the five lowest

$V_{ex}$ are almost the same as for the equal-weights method with slightly different changes. The lowest values are reported for Fischingen ($V_{ex}$ = 0.04), Bichelsee-Balterswil ($V_{ex}$ = 0.05), Wilen ()$V_{ex}$ = 0.06), Bettwiesen ($V_{ex}$ = 0.06) and Berlingen ($V_{ex}$ = 0.06). This latter region differs from the areas with lowest values for the equal weighting scheme and although having a low presence of irrigation infrastructure, it shows, among others, low share of intensive livestock, high values of farm size as well as clay rich sand as dominant soil texture.

From a comparative point of view, (Fig. 5, bottom row) results from Vex point to the northwest part of Thurgau with higher vulnerability to drought in agriculture driven by the spatial cluster from the factors distance to large water bodies, water holding capacity, and soil texture. The LAU2 regions showing higher vulnerability with $V_{ex}$ (+0.04) are mostly located between the river Rhine and Lake Constance (Homburg, Raperswilen, Herdern, Basadingen-Schlattingen, Wäldi). In contrast, regions with decreased vulnerability with $V_{ex}$ (-0.02) are located in the Northeast (Dozwil, Salmsach, Egnach) and North (Gottlieben), and

in central Thurgau (Weinfelden).

During the participatory validation with Thurgau's experts the most important factors (see Fig. 4) have been discussed regarding their correctness. The expert agreed that in general all these factors present plausible patterns across the region. However, they expected more irrigation in the 'Thur valley', the valley along the river Thur (see Fig. 1). They explained that this valley is intensively used for agricultural purposes, and therefore expected more irrigation infrastructure. As well,

they expected less clay-rich soil with clay in this area, because of the high amount of gravel in the valley, typically a sign for sandy soils. They concluded that due to the intensive use for agriculture in the Thur valley, the soil conditions could be modified due to adapted cultivation techniques. Regarding the factor soil texture, they confirmed the clay-rich, but sandy soils in 'Unterthurgau' covering the regions Diessenhofen, Schlatt, and Basadingen-Schlattingen in northwestern Thurgau, and the 'Lauche valley', the valley along the river Lauche (see Fig. 1). Further they confirmed the clay-rich soils along the

so-called 'Seerücken', a hill range up to 723 masl between Lake Constance and the river Thur (see Fig. 1, https://peter-hug.ch/lexikon/1888_bild/45_0487). Regarding the factor humus content, they questioned the low amount of topsoil organic carbon content along the Seerücken. Regarding the mapped vulnerability of agriculture to drought, the experts confirmed that both maps present reasonable differences across Thurgau with higher vulnerability in the North compared to the South. However, the experts highlighted the vulnerability map based on the expert weighting method, because of the better presentation of





known hotspot regions. In specific, the experts pointed to hotspot regions along the Seerücken, and to the hotspot Unterthurgau (see Fig. 1). The expert weighting method presents all these hotspot regions with higher vulnerability compared to the equal weighting method (see Fig. 5, Table S5).

The reasons for, as well as the differences between the hotspot regions with high vulnerability have been discussed during the interview. The experts report the Seerücken and Unterthurgau as impacted regularly in the past. However, they explained

that Unterthurgau has access to (irrigation) water from the river Rhine and from a large groundwater aquifer filled by the Lake and the Rhine. Thus drought typically leads to impacts on agriculture, if soil moisture is abnormally low. This is different regarding the Seerücken, during drought conditions typically characterised by low soil moisture, but additionally by low river discharges leading to impacts on agriculture. The experts pointed to the Lauche valley that is specifically better highlighted by the expert weighting method as the regions Stettfurt and Lommis are presented more vulnerable (+ 0.04, + 0.02) compared to

other regions (Table S5). The experts presented the Lauche as a medium-sized river delivering water for several uses that is quickly overused during drought conditions. Subsequently, in the past user conflicts on irrigation occurred regularly first in the Lauche valley, which is also the case in the current dry situation (21st June, 2022).

## 4.3   Vulnerability Podravska

Podravska's vulnerability is calculated considering 14 factors for both the equal weighting and the expert weighting method

(see Fig. 3). Within them, the four factors access to local food market, farm diversification, presence of irrigation infrastructures and soil texture increased their weights in the expert weighting scheme accounting for a total of 56 % of the overall vulnerability weight.

According to the experts, the most influential factor is the presence of irrigation infrastructure, which accounts for 15 % of the total vulnerability. Irrigated areas are mainly located in the flat areas along the Drava and Polskava rivers which are crossing

from North-West and South-West to East the Podravska region (Miklavž na Dravskem Polju, Starše, Kidričevo, Hajdina, Markovci, Gorišnica). Areas covered the least with irrigation infrastructure are mainly located in the outer parts of Podravska and at higher elevation. The second and third most important factors are soil texture and farm diversification accounting for 14 % of the total vulnerability. For soil texture we use data on the subsoil textural classes provided by the European Soil Database (ESDAC, 2022) transformed in three classes with sand as the most vulnerable texture and clay as the least vulnerable

texture according to the expert's opinion (see Table 1 1, Fig. 6). Although most of the areas in Podravska show coarser soil texture values with higher contribution to vulnerability, some clay and silt soils patches are located in the central southern area (municipalities of Slovenska Bistrica, Rače–Fram, Kidričevo, Majšperk and Makole), along the Pesnica river in the north east part of Podravska and also close to the municipalities of Ormož and Središče ob Dravi. The factor farm diversification refers to the presence of additional incomes for farmers, particularly focussing on the possibility of hosting tourists, as highlighted by

the experts. As for the factor soil texture, the factor farm diversification, shows a general condition of homogeneous low values. Only the municipalities of Maribor, Šentilj, Cerkvenjak and Ormož showed higher values in the number of permanent beds per agricultural holding (respectively of 4.86, 8.84, 7.40 and 8.55, see Table S6) which is making them less dependent on the agricultural income, hence more diversified and less vulnerable to potential drought impacts. The fourth most important factor





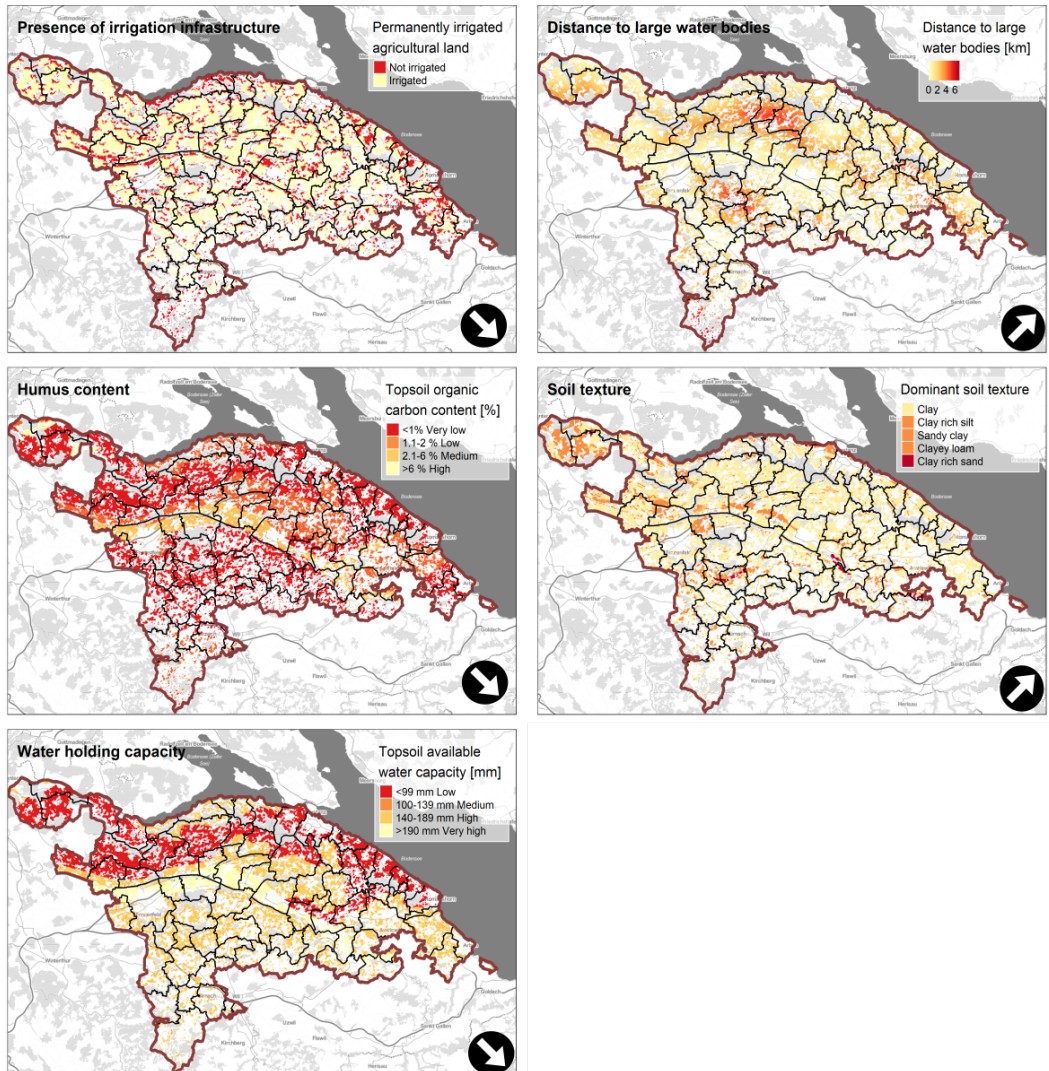

**Figure 4.** Thurgau's most important factors (bold title) masked with agricultural used land and describing the regions' vulnerability according to the experts (see Fig. 3) supported by data of the indicators (see legend). The factor's increasing or decreasing effect on the vulnerability is indicated by the arrow in the map (bottom right) and by the colour choice (the darker the colour, the higher the vulnerability). LAU2 regions are indicated by black borders and labelled when showing relatively high or low vulnerability. For an interpretation of the references to colour in this figure legend, the reader is referred to the web version of this article.

access to local food market showed a more heterogeneous situation with lower values mainly in the northern part (Šentilj)

and in the southern part (Poljčane, Makole, Žetale, Podlehnik, Cirkulane and Zavrč). The highest values were found for the municipality of Kidričevo where more than 66 % of the agricultural holdings have 'for sale' as their main destination of their products, followed by Gorišnica with 58.9 %, Rače–Fram with 58.7 % and Hoče – Slivnica with 56.8 %. The lowest values





**Figure 5.** The mapped results of equal weighting, expert weighting methods, and their difference across Podravska. On the left, masked with agricultural used land and before aggregation on LAU2 regions (black borders). On the right, after aggregation for each LAU2 region. LAU2 regions are labelled when showing relatively high or low vulnerability, or relevant differences between the weighting methods. For an interpretation of the references to colour in this figure legend, the reader is referred to the web version of this article.

were found for Podlehnik 18.8 % and Cirkulane 14.4 % in the southern part, where the agricultural holding products are mainly intended for own consumption.

Similarly to the Thurgau case study, we summed all factor raster layers and set to zero all non-agricultural areas in order to compute a mean value of vulnerability for each LAU2 region according to the equal ($V_{eq}$) and expert ($V_{ex}$) weighting methods





(Fig. 7). To identify the agricultural areas we used regional data on agriculture parcels with declared crop from the INSPIRE Slovenian Data Portal.

The five highest Veq towards agricultural drought impacts occur in the northeast part of Podravska with $V_{eq} > 0.26$. The highest $V_{eq}$ is indicated for Trnovska vas ($V_{eq} = 0.29$), followed by Sveti Jurij v Slov. goricah ($V_{eq} = 0.28$), Destrnik ($V_{eq} = 0.27$), Benedikt (Veq = 0.27) and Sveti Andraž v Slovenskih goricah ($V_{eq} = 0.26$). Moreover, other LAU2 regions with high values are still located in the northeast of Podravska. These regions show high share of agricultural areas for each LAU2 region and a combination of low values in terms of presence of irrigation infrastructure, soil texture, farm diversification and access to local food market (Fig. 6), but also for other factors with lower weight on the overall vulnerability such as distance to mountains, intensity of farming and humus content (Fig. S3).

The five lowest Veq towards agricultural drought impacts are located in the northwest and southwest parts of Podravska with $V_{eq} < 0.1$. In particular, Ruše shown the lowest Veq value ($V_{eq} = 0.03$), followed by Lovrenc na Pohorju (Veq=0.04), Selnica ob Dravi ($V_{eq} = 0.07$), Maribor ($V_{eq} = 0.08$) and Poljčane ($V_{eq} = 0.1$). These regions show a low share of agricultural areas for each LAU2 region since they are located close to mountains with forest as the main land cover type or with large urban areas as for the case of Maribor. Within these regions, the agricultural areas are located along the Drava river and benefit by the low distance to the main urban areas in terms of farm diversification.

The expert weighting method shows the highest values of vulnerability in the northeast part of Podravska with the five highest LAU2 regions being almost the same as for the equal weighting and only changing their rank as follows: Sveti Jurij v Slov. goricah ($V_{ex} = 0.35$), Benedikt ($V_{ex} = 0.32$), Trnovska vas ($V_{ex} = 0.31$), Destrnik ($V_{ex} = 0.30$) and Pesnica ($V_{ex} = 0.29$, Fig. 7b). The shift in highest values of LAU2 regions is visible in the difference between the two maps at the bottom of Fig. 7 with Sveti Jurij v Slov. goricah showed the highest increase (+0.07) driven among others by the large share of agricultural areas without the presence of irrigation, a coarse soil texture and low values of farm diversification.

The lowest values of vulnerability from the expert weighting method are also still located in the northwest and southwest part of Podravska without changes in the rank of the LAU2 regions going from Ruše ($V_{ex} = 0.04$), Lovrenc na Pohorju ($V_{ex} = 0.05$), Selnica ob Dravi ($V_{ex} = 0.1$), Maribor ($V_{ex} = 0.1$) and Poljane ($V_{ex} = 0.14$).

When comparing the expert weighting results with the equal weighting results (Fig. 7, at the bottom) there are only positive variations coming from higher weights by the experts compared to the equal weighting scheme. This means that there is a generalised worsening of vulnerability conditions as represented by the experts in Podravska. The LAU2 regions showing higher vulnerability compared to $V_{ex}$ ($\geq$ +0.05) are mostly located in the North and central South of Podravska: Sveti Jurij v Slov. goricah, Gorišnica, Kungota, Benedikt, and Starše. In contrast the regions with least changes between the two methods are spread across Podravska (Središče ob Dravi, Kidričevo, Ruše, Lovrenc na Pohorju, Sveti Andraž v Slovenskih goricah).

During the participatory validation with the experts from Podravska, we discussed each map of the most important vulnerability factors (see Fig. 6). The experts agreed that most shallow soils with coarse texture and subsequently higher vulnerability are close to rivers and springs. Therefore, they pointed to the so-called 'Drava valley' along the river Drava and the lowlands that are presented with coarse soils by the factor Soil texture in Fig. 6. Regarding the presence of irrigation infrastructure, the expert confirmed that still a large part of Podravska is not irrigated yet, which is causing problems in the North, Northeast, and



South. In contrast, they expected the East (especially Ormož) to be more covered with irrigation infrastructure, and questioned if the data source from the EEA Copernicus Land Monitoring Service published in 2018 is correctly displaying the current situation. The experts agreed on the spatial distribution of the factor farm diversification with the regions Maribor, Ormož and

Šentilj showing farms with a higher touristic share due to the presence of vineyards and orchards often offering touristic opportunities. Regarding the factor Access to local food market, they agreed that farms around cities usually have higher access to food markets, as they can sell their products, which is partly displayed around Maribor. However, they expected the East part of Podravska (e.g. Ormož) to have less access to food markets, as the markets there are known to be less developed.

Regarding the final vulnerability maps of agriculture to drought, the experts did not prefer one map out of the two methods,

as they both show the main vulnerable regions across Podravska with higher vulnerability in the Northeast, in the South and in the centre of Podravska. They pointed to the fact that subjectivity is less prominent with the equal weighting method, and that the expert weighting method should be validated in depth with local farmers working on the field. In particular, they pointed to the Drava valley and the lowlands called 'Ptuj field' and 'Drava field' (see Fig. 1), as these regions experienced drought impacts, as they are characterised by shallow soils with low water holding capacity and subsequently more vulnerable

to drought. Nevertheless, the experts agreed that the Drava valley is among the most vulnerable regions in Podravska, because of the intensive production, whereas the vulnerability maps display the North and South more vulnerable. Focusing on the Northeast and the South the experts pointed to the fact that these regions are more hilly, and additionally the water distribution is not organised in order to enable irrigation, wherefore these regions are correctly displayed with higher vulnerability. The experts agreed that the Western part of Podravska is less vulnerable, as this mountainous region is not intensively used for

agriculture, but more covered with forests. They also confirmed that the East is less vulnerable (Ormož), although more recent data should be considered to capture the local characteristics of lower access to local food market and higher values of presence of irrigation infrastructure.

## 5 Discussion

### 5.1 Sensitivity of region-specific vulnerability

Results from the vulnerability assessment highlighted how the type of weighting methods can affect each region's specific vulnerability. The implementation of different weighting methods allowed the creation of possible 'scenarios' of vulnerability conditions. Through the equal weighting we provided a neutral description of the geographical variation of vulnerability conditions. Comparably, the expert weighting method showed similar spatial patterns with a higher vulnerability. Comparing the outcome of the two different methods provided a wider perspective on potential conditions to address, an advantage acknowl-

edged by the experts during the group interview. In particular, keeping such a perspective helps to inform stakeholders and decision makers when results are particularly difficult to validate through quantitative data only.

The equal weighting method is a simple and easily computable method (Kienberger et al., 2016; Becker et al., 2014; Schneiderbauer et al., 2020) showing some limitations. In particular, the weight of each factor is dependent on the number of factors included in the analyses. This means, the more factors are considered the less weight is assigned to each factor. This reduces





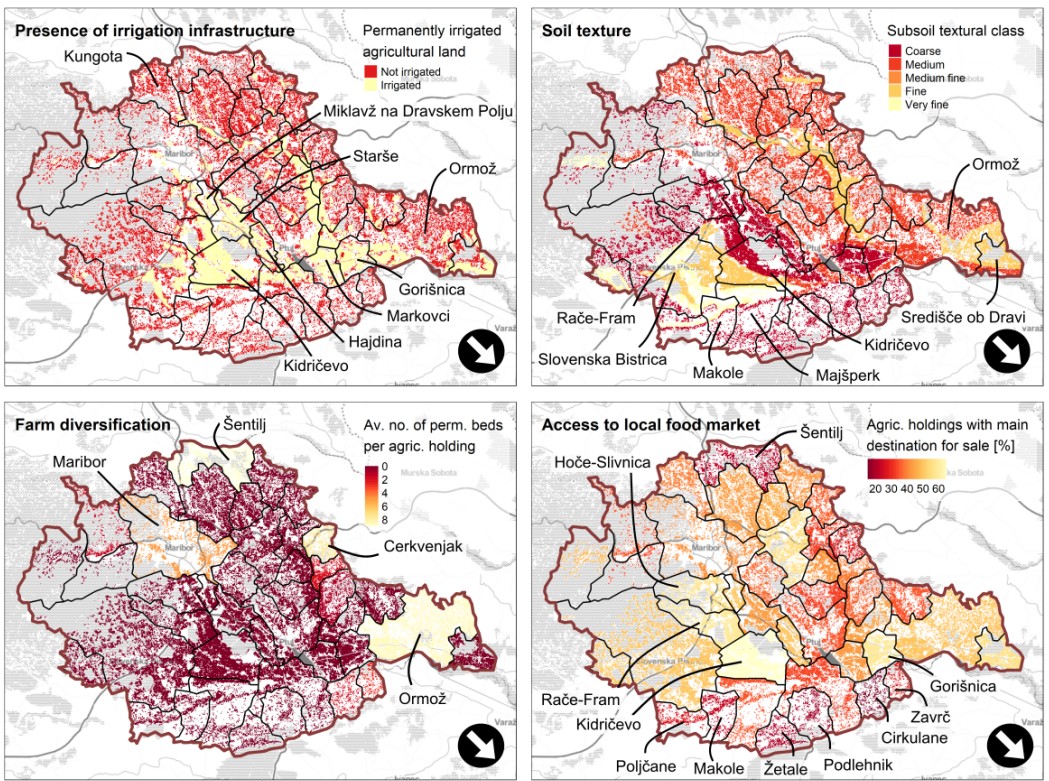

**Figure 6.** Podravskas's most important factors (bold title) masked with agricultural used land and describing the regions' vulnerability according to the experts (see Fig. 3) supported by data of the indicators (see legend). The factor's increasing or decreasing effect on the vulnerability is indicated by the arrow in the map (bottom right) and by the colour choice (the darker the colour, the higher the vulnerability). LAU2 regions are indicated by black borders and labelled when showing relatively high or low vulnerability. For an interpretation of the references to colour in this figure legend, the reader is referred to the web version of this article.

the effect of each factor on the final vulnerability map. Subsequently factors with a wider range get more influence. In our study regions, these weights were rather similar, as we could identify and supply 12 factors with data in Thurgau meaning each factor weighs $112 \approx 0.08$, and 14 factors in Podravska meaning each factor weights $114 \approx 0.07$.

The expert weighting method uses the experts' opinion built on specific knowledge of regional drought leading to agricultural impacts in the study regions. For example, the factor distance to large water bodies received substantially higher weight in both
study regions than the equal weights. Thus, the expert weighting method benefits by integrating region-specifics that might be missed by the other method. Moreover, this approach guarantees a higher level of involvement of local stakeholders (Menk et al., 2022).

Looking at the vulnerability factors identified by the experts, our study presents several common factors between both study regions. Nevertheless, the majority of the identified factors were solely mentioned for one of the regions. This demonstrates
the region-specifics and accordingly the limitation to extrapolate the expert weights or factor selection to other regions. In





**Figure 7.** The mapped results of equal weighting, expert weighting methods, and their difference across Podravska. On the left, masked with agricultural used land and before aggregation on LAU2 regions (black borders). On the right, after aggregation for each LAU2 region. LAU2 regions are labelled when showing relatively high or low vulnerability, or relevant differences between the weighting methods. For an interpretation of the references to colour in this figure legend, the reader is referred to the web version of this article.

this study, only for one out of ten common factors the assigned direction by the experts differed between the two case studies (i.e. altitudes) showing similar underlying processes leading to drought impacts. Regarding the common factors, in both study



regions the importance rating differed slightly. The factors presence of irrigation infrastructure and soil texture played a major role in both regions and, in conclusion, can be considered generally important for agriculture's vulnerability to drought in pre-Alpine regions.

## 5.2 Strengths and weaknesses of the mixed method approach

This study applied a mixed-method approach based on qualitative and quantitative information for the assessment of agriculture's vulnerability to drought. First of all, the implementation of a mixed-method approach supports a comprehensive understanding of the region-specifics, as the factors driving or mitigating the regions vulnerability have been discussed during several interviews and analysed quantitatively with the help of indicators. This way, misconceptions and false conclusions are clarified during the discussion with and between the regional experts.

One of the main limitations when performing local analysis is the availability of suitable quantitative regional and local data to support the identified factors. For this reason, the involvement and exchange of information with local experts provided a solid baseline of references and datasets for spatial and temporal data. Nevertheless, for Thurgau we were not able to find subregional data for six out of 18 factors, and for Podraska for 11 out of 25 (Fig. S1). For Thurgau this included e.g., data for the type of irrigation infrastructure, the share of drought prone and drought resistant species, presence of drought management strategies. For Podravska also e.g., food price, farmers age and education, clear landownership, agro-technical measures, and absence of drought policy. Those and more were named by the experts as factors influencing drought vulnerability. Subsequently, they are required to project and better understand potential future impacts. In addition, not all data supporting the factors has proven to be fully suitable to describe the factors highlighted by the experts. For example, the no. of beds per agricultural holding representing the factor farm diversification in Podravska simplifies the initial meaning that farms with complementary activities are less vulnerable to drought. The here selected indicators partly described the corresponding factors. They were included in the assessment as no better data was accessible and with the agreement of the experts throughout the assessment steps. Overall, the collection of local data remains a time and cost consuming process that can be supported by the application of mixed-method approaches to overcome some of this limitation.

When quantitative data was available, we applied data from different time scales. While this introduced some degree of uncertainty to map the current condition, it represented the best possible way to combine multiple information. Moreover, most of the selected factors either are static in time (e.g. altitudes) or require years to show changes at a regional level (e.g. presence of irrigation infrastructures). Moreover, the data was only accessible with different spatial scales. As the aim was to assess and map differences across the study regions, we combined data with different resolutions in a pragmatic way.

For 3 out of 26 mapped factors, we downscaled the coarse resolution to enable data aggregation. This suggests similar conditions in regions that cannot be proven, wherefore we based our final results on mean values in LAU2 regions avoiding overinterpretation. During the participatory validation the experts confirmed the need of higher-resolution vulnerability maps as a starting point for identifying and implementing more local planning and adaptation strategies.

Furthermore, we faced the typical challenge of vulnerability assessment to unify various units and value ranges in order to aggregate the data. The literature presents several methods to do so, ranging between building classes and normalising the





values (OECD and JRC, 2008). We normalised the data leading to stronger effects on the final vulnerability the greater the value range. However, this method avoids additional subjectivity on how to define the classes for each factor making them easy to interpret.

As this study tailors the vulnerability to the agricultural sector, we therefore applied an agricultural mask; respectively we integrated the land not agriculturally used as zeros in the calculation of the mean values of the LAU2-regions (Section 3.4). This way we excluded land that is not agriculturally used and considered density of agricultural land as one main aspect driving each region's vulnerability with agreement of the experts. Nevertheless, the experts in Thurgau questioned the mask in some smaller regions, as they expected slightly more agricultural used land in the mountainous region of the South and slightly less
agricultural used land along the Seerücken in northern Thurgau. The experts of both regions agreed to include the agricultural mask to enable interpretable results that are specific for the agricultural sector.

The implementation of a participatory approach throughout this study enabled a continuous exchange with local experts. At each step of the analysis this exchange provided additional information, especially on local features that cannot or are not fully described by the mapped data. In particular, qualitative narratives, as provided during the semi-structured interviews and
the group discussions, integrated and supported a better characterization of local vulnerability conditions. Besides the experts' knowledge of possible data sources, their region-specific knowledge of drought leading to agricultural impacts in the study regions substantially improved the study. In particular, the regional experts can bring their knowledge that external people might not have (as in the case of access to local food markets in Podravska and share of pastures in Thurgau) or report general assumptions that might not be valid for the specific study region.

Nevertheless, the vulnerability maps are dependent on the selected experts that participated. Regarding drought leading to agricultural impacts, the perception of important vulnerability factors might vary substantially between the interviewed persons and their background, such as politicians, water suppliers or farmers (van Duinen et al., 2015). To avoid biased ratings and the omission of important factors one option is to reach a saturation point by interviewing as many persons as possible and use a statistical measure to receive a final weight (Glaser et al., 1968; Guest et al., 2006). However, this is time-costly
and might display the current opinion and not the best knowledge based on the experience with drought conditions. Another option is to question multiple experts from different fields in order to cover specific possible perceptions (Bogner and Menz, 2009). According to this, we asked regional experts with agricultural, political and scientific background to best cover various aspects (i.e. Department of agriculture and infrastructure in rural areas, Department of environment, water construction and hydrometry, Swiss Federal Institute for Forest, Snow and Landscape Research WSL, Slovenian Environment Agency ARSO,
Chamber of Agriculture and Forestry KGZS).

Furthermore this approach guarantees the involvement of local stakeholders through a 'bottom-up and participatory appraisal' (Zebisch et al., 2021). The interaction with stakeholders increases the legitimacy of results while supporting a clear communication and bridging the gap between science and society.





### 5.3 Towards adaptation strategies to decrease vulnerability

During the vulnerability assessment, we identified a range of factors characterising vulnerability conditions to drought in agriculture in both regions (Section 4.1). Some factors represent a sensitivity to static physical terrain conditions (e.g. altitude) while others were associated with conditions that can be changed in time, such as presence of infrastructures (e.g. irrigation) or farming features (e.g. livestock presence and farm size). Within this context, it is crucial to understand what are the factors to leverage the maximum decrease in vulnerability through tailored adaptation strategies. In particular, in both regions

the involved experts weighted the presence of irrigation infrastructure as the factor with the highest importance. This result provided information at a spatial level (Fig. 4 and 6) on the areas with the highest potential to decrease vulnerability through the implementation of efficient irrigation systems (e.g. drop irrigation).

In Thurgau, humus content, water holding capacity and soil texture are the other most important factors. Their conditions might be somewhat improved by the type and extent of agricultural practices, such as diversified crop rotations, providing

organic matters to the soil (e.g. plant residues, organic fertiliser), keep the soil properly limed, and apply site-specific cultivation practices to avoid erosion and compaction (Wezel et al., 2014; Hobbs et al., 2008). While some of these practices might already be locally implemented, a regional variation of these factors' values require widespread practices implemented over years.

In Podravska, farm diversification and access to local food markets presented high potential to decrease vulnerability. In case of farm diversification, farmers could diversify their intake, for example creating or increasing their touristic offers in order to

be less dependent on the pure agricultural production. According to the experts, the owners of vineyards and orchards already offer overnight stays in apartments or touristic tours, a strategy that other farms could implement. Moreover, to increase the access to local food markets, farmers rely on the existence of food markets nearby. This is typically the case for farms close to large cities such as Maribor or Ptuj, but not in rural areas distant from urban centres. Improving road connectivity in rural areas to urban centres or developing cooperative partnerships could enable access to food markets with a higher variety of

agricultural products, as farms could combine their products and share transportation costs (Mather and Preston, 1980).

While some of these adaptation strategies are already implemented in parts of both regions, our results show which factor to prioritise and where to intervene in order to trigger the largest decrease in agricultural vulnerability to drought conditions. The presented strategies range from land-use planning, over farm cultivation techniques, towards policies that strengthen the agricultural sector to better cope with drought. Further adaptation planning should integrate them with the available local

infrastructural and financial resources through efficient communication among the stakeholders from agriculture, policy and science.

## 6 Conclusions

Drought vulnerability is defined by various regional and local factors, which differ depending on the region's conditions and on the regions' adaptation due to experiences of past drought impacts. In this study, we identified a wide range of vulnerability

factors with differences and similarities among the two case study regions highlighting the complexity of vulnerability and the difficulty to upscale the results to other, respectively larger regions, such as the whole Alpine Space. For both case study



regions we could not support all vulnerability factors with data, a restriction that can be interpreted as uncertainty of the presented results. Additionally, the different weighting methods aggregating vulnerability factors can serve as a measure of sensitivity towards the calculation approach. Being aware of the range of sensitivity and uncertainty, the final results can be

embedded better in the region specific context, which is essential for any planning and adaptation strategies. Even though this comes along with limitations in the quantitative part of the mixed-method approach, the resulting vulnerability maps were in general confirmed by the regional experts showing that we might not need to draw the complete picture of all facets defining vulnerability. Other regions may benefit from the approach presented here and tailor the participatory approach with their regional experts to validate the quantitative results with region-specific knowledge. This highlights the benefits of the mixed-

method approach combining quantitative with qualitative analyses. The results of this vulnerability assessment identified a range of adaptation strategies dependent on regional resources and efficient communication between the agricultural, political and scientific professionals. Many facets of the described adaptation strategies to decrease vulnerability to drought are in accordance with sustainability goals and climate change adaptation demonstrating the need to move from emergency actions to better preparedness. In order to better understand and quantitatively describe feedback relations and interactions between

vulnerability factors the development of drought vulnerability models integrating non-linearities is required, a field still highly underexplored.

*Author contributions.* Ruth Stephan: Conceptualization, Methodology, Data Curation, Formal analysis, Visualisation, Writing - Original Draft, Writing - Review & Editing. Stefano Terzi: Conceptualization, Methodology, Data Curation, Formal analysis, Visualisation, Writing - Original Draft, Writing - Review & Editing. Mathilde Erfurt: Conceptualization, Methodology, Formal analysis, Writing - Original Draft,

Writing - Review & Editing. Silvia Cocuccioni: Conceptualization, Methodology, Formal analysis, Writing - Original Draft, Writing - Review & Editing. Kerstin Stahl: Conceptualization, Supervision, Writing - Review & Editing. Marc Zebisch: Conceptualization, Supervision, Writing - Review & Editing.

*Competing interests.* The authors declare that they have no conflict of interest.

*Disclaimer.* This research has been supported by the EU Interreg. Alpine Space Programme project ADO (Alpine Space Observatory; grant

no. ASP940). Additional funding was provided by the Provincia autonoma di Bolzano – Alto Adige within the AquaMount project (grant no. D59C20000160003)

*Acknowledgements.* This research was funded by the EU Interreg. Alpine Space Programme project ADO (Alpine Space Observatory; grant no. ASP940). We acknowledge all the experts involved in this study for their time and support. We thank our ADO partners from Podravska



and Thurgau for their effort and support in data collection. The authors acknowledge the Provincia autonoma di Bolzano – Alto Adige

Ripartizione Innovazione, Ricerca e Università as a financing institution of the AquaMount project.



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
