# Peer review of "Assessing agriculture's vulnerability to drought in European pre-Alpine regions"

_EGUsphere, 2022_

## Author Comment (AC1)

**Reply to Referee 1**

We would like to thank you for your constructive comments and feedback on this manuscript. We think that the suggested revisions based on the Referee's comments will certainly improve the article. Please find our responses (in blue) to the main points raised (shown in black) below.

The manuscript deals with the estimation of drought vulnerability for agricultural activities in pre-Alpine climates. The authors propose a so-called "mixed method" approach which consists in taking several vulnerability factors, normalizing them and making a weighted sum of them to derive an empirical vulnerability index. The approach is "mixed" because the choice of the factors and their weighting (in the "expert weighting" method) take into account expert advice (interviews, questionnaires, …). Indeed data availability limits significantly the applicability of the approach, as the experts can recommend factors without taking into account the real availability of datasets for the factors.

→ The identification of factors through experts' interviews contributed to represent the region-specific vulnerability conditions that might not have been possible to identify using a purely top-down approach. We agree that the lack of data to represent some of the identified factors hampered a full description of the potential vulnerability. For this reason, we decided to explicitly refer to the partial availability of data representing both the initially identified and the finally selected factors in Figure 3. By doing so, we were able to answer one of the research questions in our study. This way the study reveals an estimate of potential improvement and therefore pointing to data that should be collected in order to improve other vulnerability studies in the case study.

Overall, the manuscript is well written and methodologically sound. The main issue I see is that the structure of the manuscript needs improvements (in particular the discussion section), as it is not sufficiently concise. An improvement in this sense will increase the potential impact of the manuscript, if published.

→ We appreciate the suggestion and we will revise the discussion section in a more concise way. In particular, we will shorten and restructure the discussion section inverting 5.2 with 5.1. in order to follow the order of the research questions reported at the end of the introduction in the following way:

*5.1 Strengths and weaknesses of the mixed method approach* (answering RQ1)

*5.2 Sensitivity of region-specific vulnerability* (answering RQ2)

*5.3 Towards adaptation strategies to decrease vulnerability* (providing an outlook)

**Specific comments**

L23-25 - I agree that climate change is an issue. However, I believe that one should always mention uncertainties of the climate projections which are at the basis the IPCC reports and conclusions (see e.g., https://agupubs.onlinelibrary.wiley.com/doi/full/10.1002/2017JD027463, https://nhess.copernicus.org/articles/20/3057/2020/)

→ Thanks for mentioning this point. We will refer to the climate change uncertainty providing peer reviewed articles with reference to the greater alpine and pre-alpine areas.

L204 - Perhaps explain a bit what "political conservative vote" is.

→ We will add an explanation of what the experts meant with this factor.

L224 - the factor "distance to large water bodies" may not take into account the presence of water transfers between the bodies. Is this a relevant issue for the area? Perhaps a comment on this could be added, in that case

→ We acknowledge that other factors can affect agriculture's vulnerability to drought, such as 'water transfers'. However, we only considered factors that were identified, reported and discussed by the involved experts in order to account for their knowledge on factors relevant for the regions. Nevertheless, we will better highlight this assumption in the methodology and the discussion to improve its clarity and transparency within the manuscript.

Sect. 4.1 - The vulnerability factors may be somehow statistically dependent (collinearity). I am thinking, e.g., at soil texture and water holding capacity. A comment on this is desirable. Also, how this can be prevented when involving the experts for suggesting vulnerability factors?

→ In this study, we applied the mixed-method approach letting the experts identify those factors considered as relevant according to their knowledge. By doing so, we considered all the identified factors without introducing any external assumption, e.g. on statistical correlation. This way, we addressed the first research question on the systematic identification of vulnerability factors with the support of the regional experts. Nevertheless, we see the need of mentioning the alternative approach to use statistical tests on correlation in case of quantitative data modeling and analysis and we will specify it in section 5.1 on Strengths and weaknesses of the mixed-method approach.

Sect. 5 - Discussion repeats many concepts already presented in methodology and results sections, and thus must be shortened. An important point that could be more discussed is how the approach/results can be somehow extended to other regions with different climate and socio-economic conditions.

→ Thanks for this point. We will shorten the discussion according to your suggestion. Additionally, we will elaborate a bit further on transferability and application potential to other regions.

**Minor points**

Equations are in unusual notation: the definition intervals of the variables are mixed with operators. Perhaps separate the two things as commonly done

→ Thank you for pointing this out and we will improve the equation according to your suggestion.

---

## Author Comment (AC2)

**Reply to Referee 2**

We would like to thank you for your constructive comments and feedback on this manuscript. We think that the suggested revisions based on the Referee's comments will certainly improve the article. Please find our responses (in blue) to the main points raised (shown in black) below.

The paper entitled "Assessing agriculture's vulnerability to drought in European pre-Alpine regions" aims to evaluate and understand agriculture's vulnerability to drought in two different case studies in the pre-Alpine region. The methodological approach, based on recent literature on impact chains (IC) aims at integrate quantitative and qualitative information for assessing drought risk condition. It is applied in this manuscript on two case studies in Switzerland and Slovenia. Overall, the manuscript is well written, even if:

-The results section is too much dispersive (many of the information in the text of the results paragraphs should be synthesized in tables);

→ Thanks for this point. We will revise the results section, shorten it and move information to tables where possible.

-The conclusion section is too much concise, poorly explaining the practical impacts and benefits of this approach. In the conclusion section, it would be also interesting to read future developments of this methodology in relation with the "non-linearities" the authors are referring to.

→ We appreciate the summary on limitations and prospects provided by the reviewer and will expand the conclusion with more thoughts on those aspects, especially with the focus on the benefits of the mixed-method and on how this can be applied in other regions. Further, we will elaborate more, how the approach can be improved for follow-up studies.

My main concern about this manuscript lies in the application of the methodology. Identifying Vulnerability factors through semi-structured interviews, produces highly site-specific results. Even in the usage of equal weighting method, vulnerability factors are different between two case studies. In expert weighting method, this difference is clearly more evident, highlighting that for some of the specific vulnerability factors (Figure 3), as reported (in Figure S1) unfortunately are not available information (such as regarding irrigation infrastructures).In order to built a strength methodology based on semi-structured interviews, information should be collected on a wider data sample (with numerous case studies) otherwise findings are too site-specific, as it is understandable that for their specific characteristics, the two case studies have different vulnerability factors, but without a wider comparison it is difficult to identify main common ones. This represents a weakness of the manuscript. In addition, lack of data regarding the management strategies doesn't help, as they can represent a key point for testing the methodology. From this perspective, scientific soundness of the whole manuscript should be improved, even if it represents an interesting an useful piece of knowledge specifically for both case studies.

→ Thanks for raising your concerns. We agree that a majority of the identified vulnerability factors are site-specific. In particular, this study aims to identify site-specific factors in order to evaluate and discuss agriculture's vulnerability to drought conditions in the two case study

areas. Through the involvement of the experts in the qualitative analysis (Figure 2) we were able to integrate quantitative data with their opinions and narratives on regional conditions (i.e., in Thurgau and Podravska) for interpreting and validating the final results. We acknowledge that for some factors spatial and sub-regional data was not available to describe local conditions (e.g. for drought management). However, the involved experts confirmed and supported the final vulnerability maps and their hotpots through qualitative information on the regional context (e.g., group interview for validation).

Further, we agree that having a higher number of case study regions would support the transferability of this study to broader areas. We will clarify the use of site-specific information and results in the description of the study objectives in the final part of the introduction as well as underlying it in the discussion. We can point to a broader analysis as a potential follow-up of this study to further support the transferability of this application to other pre-Alpine areas.

Some Minor Remarks:

- Figure S1 in supplement material: please, improve the quality of this figure.
- Figure S2 – S3: Text in legend is too small, please plot it bigger.

→ We will improve the quality of the Figures in the Supplementary Material.

---

## Author Response (AR1)

**Reply to Referee 1**

We would like to thank you for your constructive comments and feedback on this manuscript. We think that the suggested revisions based on the Referee's comments will certainly improve the article. Please find our responses (in blue) to the main points raised (shown in black) below.

The manuscript deals with the estimation of drought vulnerability for agricultural activities in pre-Alpine climates. The authors propose a so-called "mixed method" approach which consists in taking several vulnerability factors, normalizing them and making a weighted sum of them to derive an empirical vulnerability index. The approach is "mixed" because the choice of the factors and their weighting (in the "expert weighting" method) take into account expert advice (interviews, questionnaires, …). Indeed data availability limits significantly the applicability of the approach, as the experts can recommend factors without taking into account the real availability of datasets for the factors.

→ The identification of factors through experts' interviews contributed to represent the region-specific vulnerability conditions that might not have been possible to identify using a purely top-down approach. We agree that the lack of data to represent some of the identified factors hampered a full description of the potential vulnerability. For this reason, we decided to explicitly refer to the partial availability of data representing both the initially identified and the finally selected factors in Figure 3. By doing so, we were able to answer the first research question of our study. This way the study reveals an estimate of potential improvement and therefore points to data that should be collected (Fig. S1) in order to improve other vulnerability studies in the case study.

Overall, the manuscript is well written and methodologically sound. The main issue I see is that the structure of the manuscript needs improvements (in particular the discussion section), as it is not sufficiently concise. An improvement in this sense will increase the potential impact of the manuscript, if published.

→ We appreciate the suggestion and we revised the discussion section for more conciseness and more structure. The revised discussion follows the following sections:

*5.1 Strengths and weaknesses of the mixed method approach* (answering RQ1)

*5.2 Sensitivity of region-specific vulnerability* (answering RQ2)

*5.3 Towards adaptation and potential transferability* (providing an outlook)

**Specific comments**

L23-25 - I agree that climate change is an issue. However, I believe that one should always mention uncertainties of the climate projections which are at the basis the IPCC reports and conclusions (see e.g., https://agupubs.onlinelibrary.wiley.com/doi/full/10.1002/2017JD027463, https://nhess.copernicus.org/articles/20/3057/2020/)

→ Thank you for pointing it out. We introduced reference to the recent IPCC WGII report to complement the more general descriptions of the first sentences of the introduction and for information on the uncertainty related to future climate change conditions. As we do not consider future climate (nor any climate data at all) in this study we do not think a more specific reference to climate model signals and uncertainties for the specific region is necessary.

L204 - Perhaps explain a bit what "political conservative vote" is.

→ The factor political conservative vote was mentioned by one expert pointing to willingness to change by agricultural farmers along with drought hazards. We grouped it together with farmers' education under "background and willingness to change" in lines 210-214. We did not further consider it due to the limited data combined with the subjectivity needed to classify conservative votes into the willingness to change related to drought vulnerability.

L224 - the factor "distance to large water bodies" may not take into account the presence of water transfers between the bodies. Is this a relevant issue for the area? Perhaps a comment on this could be added, in that case

→ We acknowledge that other factors can affect agriculture's vulnerability to drought, such as 'water transfers'. However, we only considered factors that were identified, reported and discussed by the involved experts in order to account for their knowledge on factors relevant for the regions. Nevertheless, we better highlighted this assumption in the methodology (Section 3.1 l. 131-132) and in the discussion (Section 5.1) to improve its clarity and transparency within the manuscript.

Sect. 4.1 - The vulnerability factors may be somehow statistically dependent (collinearity). I am thinking, e.g., at soil texture and water holding capacity. A comment on this is desirable. Also, how this can be prevented when involving the experts for suggesting vulnerability factors?

→ In this study, we applied the mixed-method approach letting the experts identify those factors considered as relevant according to their knowledge. By doing so, we considered all the identified factors without introducing any external assumption, e.g. on statistical correlation. This way, we addressed the first research question on the systematic identification of vulnerability factors with the support of the regional experts. Nevertheless, we see the need of mentioning the alternative approach to use statistical tests on correlation in case of quantitative data modeling and analysis. Thus, we added an outlook on this in the final part of Section *5.2 Sensitivity of region-specific vulnerability.*

Sect. 5 - Discussion repeats many concepts already presented in methodology and results sections, and thus must be shortened. An important point that could be more discussed is how the approach/results can be somehow extended to other regions with different climate and socio-economic conditions.

→ Thanks for this point. We shortened the discussion and removed parts from the methods and result sections. We elaborated further on transferability and application potential to other regions at the end of the discussion and therefore changed the title of the last section to *5.3*

*Towards adaptation and potential transferability.* In addition, we integrated essential points in the conclusions.

**Minor points**

Equations are in unusual notation: the definition intervals of the variables are mixed with operators. Perhaps separate the two things as commonly done

→ Thank you for pointing this out and we updated the equations according to your suggestion.

**Reply to Referee 2**

We would like to thank you for your constructive comments and feedback on this manuscript. We think that the suggested revisions based on the Referee's comments will certainly improve the article. Please find our responses (in blue) to the main points raised (shown in black) below.

The paper entitled "Assessing agriculture's vulnerability to drought in European pre-Alpine regions" aims to evaluate and understand agriculture's vulnerability to drought in two different case studies in the pre-Alpine region. The methodological approach, based on recent literature on impact chains (IC) aims at integrate quantitative and qualitative information for assessing drought risk condition. It is applied in this manuscript on two case studies in Switzerland and Slovenia. Overall, the manuscript is well written, even if:

-The results section is too much dispersive (many of the information in the text of the results paragraphs should be synthesized in tables);

→ Thanks for this point. We revised the results section, shortened it and moved information to Tables in the Supplementary Material.

-The conclusion section is too much concise, poorly explaining the practical impacts and benefits of this approach. In the conclusion section, it would be also interesting to read future developments of this methodology in relation with the "non-linearities" the authors are referring to.

→ We appreciate the summary on limitations and prospects provided by the reviewer and expanded the discussion with more thoughts on those aspects. We elaborated with more detail, how the approach can be improved for follow-up studies (Section 5.2 and 5.3). Consequently, we changed the title of the last section in the discussion to *5.3 Towards adaptation and potential transferability* and integrated the relevant points in the conclusions.

My main concern about this manuscript lies in the application of the methodology. Identifying Vulnerability factors through semi-structured interviews, produces highly site-specific results. Even in the usage of equal weighting method, vulnerability factors are different between two case studies. In expert weighting method, this difference is clearly more evident, highlighting that for some of the specific vulnerability factors (Figure 3), as reported (in Figure S1) unfortunately are not available information (such as regarding irrigation infrastructures).In order to built a strength methodology based on semi-structured interviews, information should be collected on a wider data sample (with numerous case studies) otherwise findings are too site-specific, as it is understandable that for their specific characteristics, the two case studies have different vulnerability factors, but without a wider comparison it is difficult to identify main common ones. This represents a weakness of the manuscript. In addition, lack of data regarding the management strategies doesn't help, as they can represent a key point for testing the methodology. From this perspective, scientific soundness of the whole manuscript should be improved, even if it represents an interesting an useful piece of knowledge specifically for both case studies.

→ Thanks for raising your concerns. We agree that a majority of the identified vulnerability factors are site-specific. In particular, this study aims to identify site-specific factors in order

to evaluate and discuss agriculture's vulnerability to drought conditions in the two case study areas. Through the involvement of the experts in the qualitative analysis (Figure 2) we were able to integrate quantitative data with their opinions and narratives on regional conditions (i.e., in Thurgau and Podravska) for interpreting and validating the final results. We acknowledge that for some factors spatial and sub-regional data was not available to describe local conditions (e.g. for drought management). However, the involved experts confirmed and supported the final vulnerability maps and their hotpots through qualitative information on the regional context (e.g., group interview for validation).

Further, we agree that having a higher number of case study regions would support the transferability of this study to broader areas. We clarified the use of site-specific information and results in the description of the study objectives in the final part of the introduction (line 52-58 and line 73-82). We further underlined that it in the discussion (final part of 5.3) and pointed to a broader analysis as a potential follow-up of this study to further support the transferability of this application to other areas characterized with broadly similar conditions. As mentioned above, we therefore changed the title of the last section in the discussion to *5.3 Towards adaptation and potential transferability* and integrated relevant points in the conclusions

Some Minor Remarks:

- Figure S1 in supplement material: please, improve the quality of this figure.
- Figure S2 – S3: Text in legend is too small, please plot it bigger.

→ We improved the quality of the Figures in the Supplementary Material.

---

## Author Response (AR2)

Dear Brunella Bonaccorso,

Thank your very much for handling the constructive review process by NHESS. According to Referee #1, I removed all "." (multiplication symbols) before the variables in the equations, re-read the manuscript and removed small typing errors. Since the wording "political conservative vote" was mentioned by one expert from Thurgau, we want to use this original phrasing (as we did for all factors). According to the previous comment of Referee #1, we grouped this factor to "willingness to change" (l. 213) in order to clarify better what the expert described during the interview.

On behalf of all co-authors,

Yours sincerely,

Ruth Stephan